# Donor and Donation Images (DDI)—A Scoping Review of What We Know and What We Don’t

**DOI:** 10.3390/jcm12030952

**Published:** 2023-01-26

**Authors:** Nora M. Laskowski, Gerrit Brandt, Katharina Tigges-Limmer, Georg Halbeisen, Georgios Paslakis

**Affiliations:** 1University Clinic for Psychosomatic Medicine and Psychotherapy, Campus East-Westphalia, Medical Faculty, Ruhr-University Bochum, Virchowstr. 65, 32312 Luebbecke, Germany; 2Clinic for Thoracic and Cardiovascular Surgery, Herz- und Diabeteszentrum NRW, Ruhr-University Bochum, 32545 Bad Oeynhausen, Germany

**Keywords:** donor image, donation image, organ integration, organ transplantation, heart transplantation, lung transplantation, kidney transplantation, DDI, psychocardiology, scoping review

## Abstract

Organ transplantation is associated with significant physical and psychological burden for the recipients. Qualitative reports indicate that organ recipients develop donor and donation images (DDI)—conceptions of the donor and/or the organ. A deeper understanding of DDI is needed in the care of transplant recipients. To present the current state of knowledge, we searched for and identified DDI-related publications in PubMed and Scopus. Inclusion criteria were (1) studies addressing transplant recipients, and (2) English or German language. Twenty-one studies of individuals with transplanted hearts, lungs, or kidneys were included in this scoping review. Prevalence for DDI ranged from 6% to 52.3%. DDI occurs both before and after transplantation and includes ideas about the donor as well as whether and how the recipient’s personality may be altered by the transplanted organ. Some transplant recipients did indeed report personality changes following transplantation due to the adoption of assumed donor characteristics. One study showed a positive association between the presence of DDI and anxiety scores and one described a coping effect. DDI is understudied and should be systematically assessed to improve care for the vulnerable group of individuals undergoing organ transplantation. Current research gaps and future directions are discussed.

## 1. Introduction

Organ transplantation has become a standard treatment for organ failure, serious injury, or irreversible disease and may prolong the lives of critically ill people [1]. On the other hand, it is a serious medical intervention and a possibly life-threatening event itself with far-reaching consequences for the individuals involved. There is research on transplantation-associated psychological factors of organ donors (see e.g., [2,3,4,5,6]) and on psychological factors associated with the procedure on the organ recipients’ side as well (see e.g., [7,8,9,10]). Studies show that, in terms of psychological distress, the most frequently reported symptoms among organ recipients are anxiety symptoms (particularly fear of organ rejection, repeated surgery, and death) and symptoms of depression [11,12,13]. In addition, up to 16% of individuals show post-traumatic stress disorder (PTSD) following transplantation with symptoms such as scenic re-experiencing of events, emotional overload, nightmares, high internal tension, withdrawal, and avoidance behavior [14,15]. Symptoms of this kind are due to the mental/psychological burden before, during, and after transplantation (e.g., due to treatment in the intensive care unit) accompanied by a continuously experienced threat of death. These result in a reported low quality of life [16]. Psychological distress may have serious consequences for the recovery process and thus a negative impact on the transplantation’s outcome [1,17,18]. The presence of depression, for instance, has been associated with the loss of the transplanted organ and increased mortality [11]. For heart transplant patients in particular, depression is associated with increased mortality in the long-term course [19]. Therefore, it is important to identify potential psychological stressors, especially since the determinants of a successful psychological and physical integration of transplanted organs have not yet been fully understood [1].

Qualitative reports from organ recipients point towards a possible source of psychological distress lying in (mis-)conceptions developing around the organ donor or the received organ itself [1,20]. This phenomenon is frequently described as “fantasies about the donor” (e.g., [21]) or “fantasies about the organ” (e.g., [22]). While a generally accepted definition of the phenomenon is still lacking, we propose the term “donor and donation images” (DDI) to avoid the stigmatizing undertone of the term “fantasy” (describing a sham entity). DDI include all (magical and/or rational) thoughts and feelings that an organ recipient has associated with the donor or the donated organ itself and are often accompanied by a change or fear of change in personality traits in individuals with transplantation experience attributed to the transplantation [23]. As psychological and physiological rejection responses are partly interrelated [17], a deeper understanding of DDI (e.g., time of occurrence, characterization) is relevant to the care of post-transplant individuals and therefore highly needed.

The evidence on DDI is scarce [1]. Existing reviews include DDI only as a marginal topic, search approach and result presentation are unsystematic or they only refer to single organs (e.g., [20,24,25]). There has been narrative evidence of DDI in the literature for 20 years, but to date, no systematic assessment tools have been developed, neither clinical implications been explored. This scoping review aims to present the current state of knowledge on DDI. To systematically summarize the existing knowledge on DDI, we present information regarding (1) terms used in connection with DDI, (2) prevalence, (3) time of occurrence (related to transplantation), (4) specific contents, and (5) consequences. Results are related to different organs (heart, lung, kidney). Current research gaps and future directions are discussed.

## 2. Materials and Methods

No pre-registration was performed. We carried out a literature search using the databases PubMed and Scopus; the databases were last consulted on 10 October 2022. Studies were eligible for inclusion according to two criteria: first, studies needed to address organ transplantation (regardless of the organ). Second, studies needed to be published in English or German language.

The search string (see Table 1) incorporated terms mentioned in articles and was thus constantly expanded. Search terms needed to be indicated in titles or abstracts.

Publications were selected using a two-step procedure. In the first step, titles and abstracts were screened, and additional screening of the reference lists in the identified publications was performed by the first author. Ambiguities were discussed and resolved among all authors. In the second step, the full-text review was performed by the authors NML, GB, and GH in random selection. In the case of further doubts regarding exclusion or inclusion, a consensus was reached with the assistance of the other authors. The final selection of studies based on full-text reviews was approved by all authors. The narrative outcomes were performed by the authors NML, GB, and GH. For this purpose, all included articles were read and all descriptions concerning DDI were summarized per organ category to provide an overview of the current state of research.

## 3. Results

Cross-sectional surveys, interview studies, and case reports were identified and included. No prospective studies were found. The following presents the information about DDI taken from the literature, separately for each type of transplanted organ. The following aspects are presented: (1) used terms, (2) prevalence, (3) time of occurrence (in relation to transplantation), (4) specific contents, and (5) consequences.

### 3.1. Inclusion and Exclusion Process

We identified a final total of 21 publications by means of database search and additional reference list screening as shown in the flow diagram in Figure 1.

Most included publications were from Switzerland (*n* = 6). The majority of publications were interview studies (*n* = 16) with rather a low number of participants (ranging from *n* = 14 to up to *n* = 47). We identified only two surveys that included *N* = 76 and *n* = 684 participants, respectively. No randomized controlled trials could be found. Most studies addressed the heart (*n* = 9), as can be seen in Table 2.

### 3.2. Heart Transplantation

Neuroscience has brought the brain into focus as the seat of psychological experience in recent years. In the belief system of many people across many cultures it is the heart, however, which represents the seat of personality traits and emotions (e.g., [23]); this is reflected in the common language phrases such as “a kind-hearted person”, “pour one’s heart out”, “heartbroken”, or “heartfelt” [43]. Linguistic metaphors of this kind can be found in many different languages. The heartbeat is unbreachable connected to “being alive” [22] and a body without a (functioning) heart may be considered a “zombie” [44]. Heart transplantation may interfere deeply with the perception and maintenance of body integrity [1].

Our search revealed nine qualitative studies addressing the phenomenon of DDI in heart transplantation; samples ranged from *N* = 14 to *N* = 47 participants.

#### 3.2.1. Used Terms

In the context of heart transplantation and DDI, terms such as “fantasies” [17,37,41,42], “incorporated fantasies” [23], “desire to know more about the donor” [40], “thoughts about the donor” [35], and “fantasies about the new heart” [22] were found.

#### 3.2.2. Prevalence

In most studies, no information is given about the prevalence of DDI in heart transplantation, while two studies made statements in this regard. Bunzel et al. [23], however, found instances of DDI in 6% of *N* = 47 patients after heart transplantation. Ivarsson et al. [35] found that “most” of *N* = 16 patients had DDI without further specifications. Kaba et al. [36] showed that some patients actively avoided thoughts and feelings related to the received heart or the donor and seemed to deny the effects associated with the transplanted organ. Individuals who actively avoid DDI seem to perceive the heart more as a mechanical entity that can be more easily replaced [36].

#### 3.2.3. Time of Occurrence

Three publications ([20,35,44]) described the time of occurrence of DDI as being post-transplantation. The studies by Stolf and Sadala [42] and Kuhn et al. [37] reported that DDI may occur anytime between the initial waiting period to years after heart transplantation. Two studies observed that pre-transplant thoughts may occur but then tend to include thoughts regarding the circumstances of the death of the donor or the general idea of receiving a heart from a deceased person [22,37]. Thoughts about losing one’s own heart were addressed by two studies [37,42]; in these, such thoughts occurred after transplantation.

#### 3.2.4. Content of DDI

Regarding the content of DDI described in the literature, several studies mentioned that thoughts about the donors’ sex and age were important aspects of DDI [23,35,36,37,40,41,42]. There are also reports that some recipients developed ideas about the donor’s personality [17], as donors are considered central figures in the process of organ transplantation [22]. Some individuals assumed that the donor (despite being anonymous) had similar traits to them [41]. Other studies reported a transfer of donors’ traits onto recipients [17,41]. Three among fourty-seven interviewed patients described by Bunzel et al. [23], for instance, believed that they had incorporated traits of the donor, and thus reported personality changes caused by the transplanted heart itself. More precisely, they reported developing new character traits, hobbies, and passions as a result of the donors’ characteristics. Furthermore, patients reported that the donor would still be alive inside their body and that they would feel like living two lives [23].

In the context of thoughts around the donors’ death, individuals have been reported to fantasize about the circumstances leading to the death of others (e.g., having an accident, “donor weather” [37]).

Some patients also reported resenting the loss of their own hearts: one patient in Stolf and Sadala’s study [42] reported that he would always remember that he was not living with his own heart. Thoughts about losing one’s heart were also hinted at in the study by Kuhn et al. [37]: a patient was quoted to have said that he/she would be leaving his/her heart in the hospital upon discharge. However, there was no further discussion of this or its impact.

#### 3.2.5. Consequences of DDI

Some emotional consequences of DDI are described in the context of thoughts regarding the donors’ death. Guilt related to the donor’s death is a common theme [17,22,36,37]; additionally, the ambiguity of gratitude and guilt has been reported [17,41,42].

In Canada, it is common to encourage transplant recipients to write a “thank you” letter to the donor’s family. In such situations, the recipients’ involvement with the donor’s life is further encouraged. In the qualitative interview study by Poole et al. [40], a strong negative emotional response to this invitation (“obligation” [40]) to compose a letter was described in a group of 27 heart transplant recipients. All of the 19 participants who wrote a letter found the process of writing to be highly distressing. Five participants were unable to write a letter due to exhibited distress.

The heart’s symbolic association with “vitality” and “soul” makes it particularly difficult for recipients to integrate a transplanted heart and potentially complicates the recipients’ acceptance of the organ [17]. The descriptions of some heart transplant recipients suggest that there may be relevant problems with organ acceptance due to the idea of the heart as the center housing emotions and shaping personality [23]. The replacement of one’s own heart by the heart of another person may become a threat to one’s well-established self-concept [17].

### 3.3. Lung Transplantation

Lung transplantation is a treatment option for various forms of end-stage lung diseases when alternative treatment options are no longer effective. Lung trabsplantation can improve the quality of life significantly [45]. Survival rates are 82% 12 months after transplantation and 70% after 5 years (conditional on survival to 1 year) with an increasing trend over the last decades [46].

Our search revealed six qualitative studies (samples ranged from *N* = 1 to *N* = 38) and 1 survey (*N* = 76) on DDI in lung transplantation.

#### 3.3.1. Used Terms

In the context of lung transplantation, the following DDI-related terms were found: “fantasy” [31] and “fantasies around the transplant experience” [29], general “(frequent) thoughts about the donor” [32,33,35], or “imagined organ donors” and an “imagined relationship with the donor” in the studies by Goetzmann et al. [30,32].

#### 3.3.2. Prevalence

The frequency of DDI in lung transplant recipients was addressed in two of the seven studies included [32,39]. In the qualitative interview study by Neukom et al. [39], six out of 20 patients (30%) reported DDI. In the cross-sectional questionnaire studies by Goetzmann et al. [32], 32.9% (of *N* = 76) respondents reported frequent thoughts regarding their organ donor. In contrast, Ivarsson et al. [35] stated that most recipients denied having DDI, without further specification.

#### 3.3.3. Time of Occurrence

Five studies described that DDI occurred mainly post-transplantation [29,31,32,35,39], with time periods after transplantation ranging from shortly after transplantation [35] to one [39] or four [31] years after lung transplantation.

The study by Goetzmann et al. [32] also reported that DDI occurred during the patients’ preparation phase for lung transplantation.

#### 3.3.4. Content of DDI

In their interview study, Eichenlaub et al. [29] described “unconscious fantasies” [29] that appeared as dreams. One patient, for instance, reported fantasies about the donor’s death: She dreamt of a violent death, which led her to describe transplantation as cannibalism regarding the incorporation of organs [29].

Some studies described that the donor was experienced as living on in the recipient [29,31]. Goetzmann et al. [33] stated that recipients in their interview study imagined their own bodies to be vessels in which the donor still existed -in the form of thoughts, but also as a tangible being. In another study, Goetzmann [31] described a single lung transplant recipient (out of a larger sample of 20) who perceived the donor as still being alive; when prompted by the interviewer, this recipient characterized the donor with attributes similar to him-/herself and reported experiencing gratitude and “talking” to the donor to cope with daily problems. Eichenlaub et al. [29] reported that 3.83% of recipients described the transplanted lung as being part of their own bodies and that they had also picked up character traits of the donor, e.g., frequent crying, which was attributed to the transplanted lung. There have also been other studies in which recipients reported that characteristics of the donor had been transferred to the recipient [29,31,33]. In the survey by Goetzmann et al. [32], 7.9% (*n* = 6) reported that they felt they had adopted the characteristics of the donor (without further specification), while 11.2% were not quite sure.

In contrast, Ivarsson et al. [35] stated that most patients denied having DDI. This information was not assessed by means of standardized survey instruments but emerged during semi-structured interviews. However, those who reported about having DDI also reportedly thought about features such as the age or sex of their donor. One study described a patient who stated that he/she had formed a very specific inner image of the donor [33].

Some authors discussed the adoption of character traits as a result of projecting one’s own traits onto the donor’s representation [32], while Eichenlaub et al. [29] concluded that the activation of unconscious fantasies seems to be a typical psychological process in lung transplant patients.

#### 3.3.5. Consequences of DDI

Goetzman et al. [32] merged all aspects regarding thoughts about the donor and the adoption of character traits, but also the fact that events accompanying transplantation no longer play a role in one factor called “relationship to the donor” [32]. This factor was shown to be a predictor of chronic stress, psychological distress, concern regarding the transplant, and guilt [32]. A different impression is given by the study by Neukom et al. [39], in which the authors described ambiguous emotions ranging from affection and gratitude to sadness and shame occurring in recipients dealing with the donor and their death [39].

### 3.4. Kidney Transplantation

In contrast to heart and lung transplantation, living organ donations from relatives or strangers are possible in kidney transplantation, an aspect that influences the dynamics of the donor–recipient relationship [27]. In the study by Kärrfelt et al. [21], the donor–recipient relationship (mostly close relatives) appeared to improve through the common experience; the hypothetical idea of having a necro-transplant was evaluated negatively [21]. Basch [26] maintained that deceased individuals as donors open up the possibility that DDI or preconceived attitudes and feelings are transferred to the donor or its organ, while other authors found out that there is also a notion in living donation to adopt characteristics of a donor through transplantation [38].

Our search revealed seven qualitative studies (samples ranged from *N* = 2 to *N* = 55) and 1 survey (*N* = 684) on DDI in kidney transplantation.

#### 3.4.1. Used Terms

Regarding terminology used in the context of DDI, almost all studies in kidney transplantation used the term “fantasies” [21,26,27,28,38,41]. Basch [26] and Kärrfelt et al. [21] generally described “fantasies about the donor”, whereas Delmar-McClure [28] used the term “fantasies about the donor’s organ”, and Crombez and Lefebvre [27] wrote about “fantasies concerning the transplantation”.

#### 3.4.2. Prevalence

Two studies addressing DDI in the context of kidney transplantation found a prevalence of DDI in kidney transplant recipients [21,34]. In the qualitative study of teenagers by Kärrfelt et al. [21], three out of 20 recipients (15%) who had received an organ as a child reported DDI. Hennemann et al. [34] found a higher prevalence: In their survey, 52.3% reported that they were frequently preoccupied with thoughts regarding the donor.

#### 3.4.3. Time of Occurrence

Regarding kidney transplantation, the time of occurrence of DDI is heterogeneous. According to Sanner [41] DDI occurred in the first six months following transplantation, Kärrfeld et al. [21] reported a range between one and twelve years after transplantation.

One study stated the occurrence of DDI before transplantation [28].

#### 3.4.4. Content of DDI

Sanner [41] reported that most recipients preferred to avoid thinking about the donor. Especially recipients of necro-transplants were identified as suppressing/avoiding thinking about the donor. However, there were also cases of necro-transplant recipients thinking about specific features of the donor, with a few reporting that they believed (or feared) that donor traits would get transferred and affect their behavior. The relevant features here were aspects of gender, whereas age played a minor role [41]. Other authors stated that sexual orientation also played a role [26,38]. Hennemann et al. [34] found that 13.6% of patients had the impression they adopted traits of the donor.

#### 3.4.5. Consequences of DDI

As a very direct consequence, Hennemann et al. [34] pointed out that thinking about the donor and beliefs about the transfer of traits were positively associated with patient anxiety scores [34].

Crombez and Lefebvre [27] reported that distorted fantasies on the recipients’ side may lead to a disruption in the relationship between donor and recipient, causing complications for both. For instant, the recipient may wish to have a donor who is different from the intended donor [27]. Two other studies reported that thoughts might depend on the recipient’s attitudes and values; if recipients know or imagine that the organ is from a donor whose gender or age group is different from their own, special problems may occur [26,38]. Recipients may endow the transplanted organ, in this case, the kidney, with the sexual or gender characteristics of the donor. The difficulties with cross-gender donations or donations from a person with a different sexual orientation may also arise with living donations [26,38]. In the case study by Delmar-McClure with two recipients [28], a male individual was identified as having a strong aversion to receiving an organ from a female donor. He stated that only a male kidney would be “strong enough” to be a good fit; this recipient attempted to commit suicide one year following transplantation. The other study participant, who was interviewed during the waiting period, viewed the recovery of the cadaveric donor organ as “grave robbing”, believing the spirit of the “robbed” person would haunt her, which in this case led to the rejection of the life-saving operation [28].

## 4. Discussion

The present scoping review aimed at summarizing the existing evidence related to the phenomenon of donor and donation images (DDI) in transplant recipients. DDI include thoughts and feelings that transplant recipients may develop about their donors and the received organs. Our search for relevant publications identified 21 qualitative studies (interviews, case reports, and two surveys) addressing the prevalence, time of occurrence, content, and consequences of DDI for organ recipients. We found no prospective or randomized controlled studies. We found publications on heart, lung, and kidney transplant recipients. While DDI is our preferred term, others have used terms such as “fantasies”, “thoughts about the donor”, or “imagined relationship with the donor”. All in all, our extensive search reveals that DDI are an understudied phenomenon and results can only be narrative in nature up to this point.

Reports regarding the prevalence of DDI across the different groups of transplant recipients (heart, lung, kidney) differed significantly among publications, most publications did not assess or provide prevalence data though. In the publications that did provide numbers, DDI were very common in some groups and less common in others, with rates ranging from 6% [23] to 52.3% [34] of recipients. The broad prevalence rates might indicate that boundary conditions and triggering aspects, so far largely unknown, rather than specific time windows may best predict the emergence of DDI. Investigations taking influencing variables into account, including longitudinal designs, are therefore necessary. One particular group of transplant recipients seems to actively avoid getting involved in DDI. Again, no reliable rates for this group can be presented based on the existing literature. Active avoidance of DDI-associated expressions may be understood as a coping mechanism. Confrontation with the donors’ next of kin (e.g., composing thank-you-letters to the bereaved) has been reported to induce a negative emotional reaction in transplant recipients [40]. In Germany recipients can (mediated by the German Foundation for Organ Transplantation) write anonymous letter of thank-you to the donor ‘s family, though no instance of this has come up in any of the studies. In our clinical experience, the voluntary and anonymous nature of the letter appears to be relevant for its mediation of relief.

In general, as regulations regarding the disclosure of information about donors differ significantly between countries (e.g., prohibition of disclosure in Germany), formal legal aspects may influence observations by impacting individuals undergoing heart transplantation in a different manner in different countries; no reliable evidence on these aspects can be provided now.

The point of time when DDI occurred was described to lie after transplantation in most of the publications (up to 12 years post-transplantation [21]). Only four publications reported that DDI also occurred during the waiting period pre-transplantation [22,28,32,37]; apparently, pre-transplantation DDI revolve around the donors’ circumstances of death.

We found publications describing DDI of a magical character, such as the absence of a heart making a “zombie” [23] out of someone, or the idea that donor traits may be passed over to the recipient by means of the transplanted organ [37], a sexist attitude that only a male organ could fit a male patient was also mentioned [28]. It appears likely that DDI do not arise independently of predominant social narratives and myths and misconceptions regarding organ function. It is also somewhat surprising that the content of DDI has not significantly changed from the older studies (in the 70es, 80es, and 90es) to the more recent ones, as the former also reported that character traits might be transferred onto transplant recipients (e.g., [23,32,34]); one would expect magical beliefs to decline as societies become better educated and informed. Physical memory processes, such as cardiac memory, might explain part of the modified self-perception in patients following heart transplantation. For instance, alterations in physiology such as regarding heart rhythm, stored in the donor heart (“remembered”), might bring about lingering changes in recipients’ physical perception of the organ [47], although this has to be considered highly speculative at this point. Nonetheless, we would guess that organ transplantation constitutes an exceptional human experience that might also activate exceptional psychological mechanisms. The psychological mechanisms underlying DDI need to be explored in corresponding studies. Besides, aspects of gender, age, and even sexual orientation of the donor described as prevalent in the literature will have to be taken into account.

According to what is known today, the temporal development of DDI and DDI-associated effects on transplantation outcomes are unknown. DDI may be experienced as burdensome as they are highly emotionally activating (in terms of gratitude, concerns, regret, guilt, self-rejection, disgust, and/or feelings of ambivalence) but their influence on healing and the organ acceptance process is largely uninvestigated. The type of questions used in the interview studies may have caused a bias (e.g., in terms of the emotional response of transplant recipients when referring to a “deceased” donor). This makes it all the more important to use standardized questionnaires. A single study reported an association between the presence of DDI and anxiety scores [34], while another report described DDI as helpful in terms of emotional coping [31] (e.g., an imagined donor with whom daily problems are discussed or behavioral change attributed to features of the donor). In clinical practice, the relatives of transplant recipients report their own DDI. The extent to which DDI of the next of kin are similar to those of the recipients or their influence on the transplant recipients’ health condition is completely unexplored up till now.

A limitation of the scoping review process is that the literature search was conducted by only one person. Articles were selected by the authors NML, GB, and GH–uncertainties were resolved by the entire study team. In terms of content assessment, a few limitations preclude drawing reliable conclusions so far. First, as mentioned above, irrespective of the methodological approach, the construct of and the terms used to refer to the phenomenon of DDI have insufficiently been defined to date. That is why any assessment of the validity of the construct is rendered rather difficult. It is still unclear, whether DDI are a specific form of rumination, a coping mechanism, or are characterized by any unique properties unrelated to general imaginative tendencies. The epistemological status of DDI thus still needs to be established. Moreover, we do not know whether and how DDI change or remain stable over the course of the transplantation process. Second, while smaller samples are common in qualitative interview studies, it is necessary to refer to high-powered quantitative studies for robust conclusions–these are lacking. Systematic surveys on DDI are scarce as well, with only one kidney-related [34] and one lung-related publication [32] included in this scoping review. Consequently, even if we assumed that the existing publications explored the same phenomenon, we would have to note that the qualitative approach undertaken in most cases limits any conclusions about specific prevalence rates, times of occurrence, contents, and consequences. Third, heart and lung transplantations are necessarily accompanied by the donor’s death, while kidney transplantations include living donor transplants. Additionally, while donors are unknown/anonymous in the case of heart and lung transplantation, donors are often affiliated with kidney recipients. Looking at differences in DDI and their psychological reflection between living donations and donations from deceased individuals represents an important topic for further studies. The specific differences between procedures may account for differences in prevalence, the time point of occurrence, content, and consequences of DDI but current research data do not allow for definite conclusions in this respect.

We found no study on DDI in liver transplant recipients. A possible explanation for this could be that organs such as the heart and the lungs are associated with body functions that are not only at all times easy to notice and observe at all times, but also strongly associated with vitality and being alive itself (pulse/heartbeat, breathing). Forinstance, as a result of heart transplantation, patients start to perceive the strong heartbeat of the new organ compared to the previously weak heartbeat of their ill heart. Such organ-specific aspects could predispose for the development of DDI. In addition, studies on liver transplantations have mostly focused on adherence to treatment, alcohol abstinence, etc. rather than DDI. Future studies could cast more light onto differences between organs working “silently” and those associated with obvious and palpable, etc. features.

Finally, the relationship between DDI and psychological distress has not been studied sufficiently, even though there are some reports on the emotional activation due to DDI. There is no evidence as to what degree DDI represent emotional distress that may be detrimental to health outcomes related or unrelated to the transplantation itself. DDI may just as well co-occur with general health or organ integration impairments rather than be their cause. Still, as psychological and physiological rejection responses are partly interrelated [17], a deeper understanding of DDI is needed and highly relevant to the care of transplant recipients. The relationship between DDI and adherence to treatment would also need to be investigated. As a tentative outline for future research, we suggest that studies should first describe the contents and structure of DDI, assess their prevalence and health consequences quantitatively, and thus establish whether addressing DDI in transplant recipients may promise therapeutic potential.

## Figures and Tables

**Figure 1 jcm-12-00952-f001:**
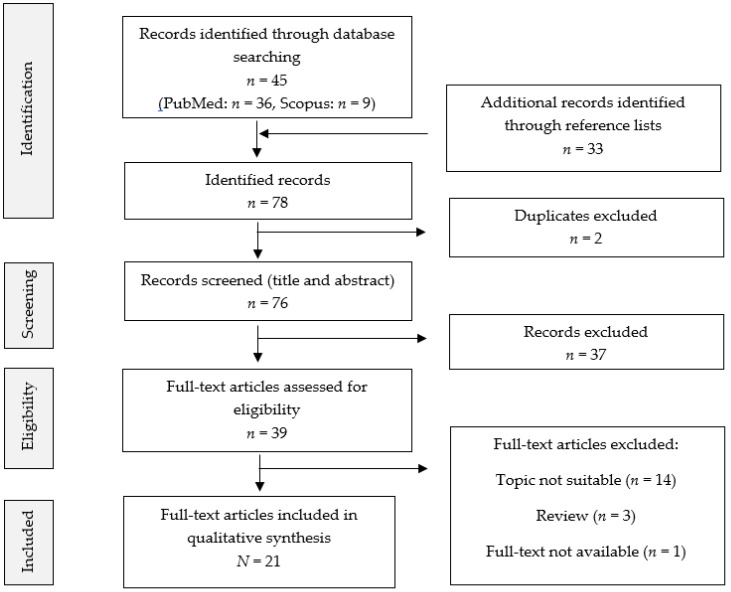
Flow diagram showing the inclusion and exclusion process.

**Table 1 jcm-12-00952-t001:** Search string used for the search in pubmed.gov.

Search String [Abstract/Title]
((incorporation fantas*[Title/Abstract]) OR (organphantas*[Title/Abstract]) OR (organfantas*[Title/Abstract]) OR (fantas* traits of the don*[Title/Abstract]) OR (fantas* about the donor[Title/Abstract]) OR (Spenderphantas*[Title/Abstract]) OR (Spenderfantas*[Title/Abstract]) OR (fantas* about the organ[Title/Abstract]) OR (Fantas* recipient-donor relationship[Title/Abstract]) OR (donor fantas*[Title/Abstract]) OR (thoughts about the donor[Title/Abstract]) OR (fantas* about the donor’s organ[Title/Abstract]))

**Table 2 jcm-12-00952-t002:** Overview of included studies.

First Author	Year	Country	Organ	*N* (*n* Women)	Design
Basch [26]	1973	USA	Kidney	28 (n. a.)	Interview study
Bunzel [23]	1992	Austria	Heart	47 (2)	Interview study
Crombez [27]	1972	Canada	Kidney	15 (n. a.)	Interview study
Delmar-McClure [28]	1985	USA	Kidney	2 (1)	Case report
Eichenlaub [29]	2021	Switzerland	Lung	38 (18)	Interview study
Goetzmann [30] ^1^	2007	Switzerland	Lung	14 (6)	Interview study
Goetzmann [31]	2004	Switzerland	Lung	1 (0)	Case report
Goetzmann [32]	2009	Switzerland	Lung	76 (33)	Survey
Goetzmann [33] ^1^	2009	Switzerland	Lung	14 (6)	Interview study
Hennemann [34]	2021	Germany	Kidney	684 (283)	Survey
Ivarsson [35]	2012	Sweden	Heart, Lung	16 (9)	Interview study
Kaba [17] ^2^	2005	Scotland	Heart	42 (7)	Interview study
Kaba [36] ^2^	2000	Scotland	Heart	42 (7)	Interview study
Kärrfelt [21]	2003	Sweden	Kidney	20 (9)	Interview study
Kuhn [37]	1988	USA	Heart	6 ^3^ (n. a.)	Case reports
Lefebvre [38]	1973	Canada	Kidney	20 (n. a.)	Case reports
Neukom [39]	2012	Switzerland	Lung	20 (10)	Interview study
Poole [40]	2011	Canada	Heart	25 (8)	Interview study
Sadala [22]	2007	Brazil	Heart	26 (n. a.)	Interview study
Sanner [41]	2003	Sweden	Kidney, Heart	35 (14)	Interview study
Stolf [42]	2006	Brazil	Heart	26 (6)	Interview study

n. a. = not available; ^1^ = Publications based on one sample; ^2^ = Publications based on one sample; ^3^ = The authors mentioned having observed a total of 65 patients but describe six cases in their article.

## Data Availability

Not applicable.

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
