# Peer review of "Donor and Donation Images (DDI)—A Scoping Review of What We Know and What We Don’t"

_jcm, 2023, doi:10.3390/jcm12030952_

Round 1

Reviewer 1 Report

I congratulate the authors on their topic. When we exclude the articles from 2021 in the included studies, we can see that these evaluations are made at intervals of more than ten years in the literature. Despite this time difference, their collection of such a critical subject shows the manuscript's importance. 

My minor comments:

*Inclusion and Exclusion criteria are well defined. 

*Limitations in literature and the lack of proper analysis are well compared.

About Heart transplantation

*Developing new traits and resenting the loss of their own heart must have caused an emotional wave. However, according to the manuscript, most DDI observations start post-transplant. Did indignation about a possible loss of their own heart start pre-operatively? If possible, please indicate. 

*All emotional processes related to the heart may be related to the patients' empathy feelings and the complex process they experience. In addition, although "cardiac memory" is a dilemma, it is still a current issue. Is there a justification for the acquired characteristic trait for cardiac memory? The manuscript mentioned the emotional adjustment process in the Kuhn 1988 study. However, brief information about cardiac memory can be added to the conclusion to increase the article's impact. (such as https://doi.org/10.1016/S0008-6363(98)00208-9).

About Lung Transplantation

*The "a deceased donor" term influences many recipients' emotions about the donor. As reported in lung transplantation, regret reveals a complex focus of stress, such as refusal of possible emotional attachment to the donor (denial).  Is a standard mood assessment test preferred in the interview studies mentioned here? Was a standard method preferred for such comprehensive evaluations in those studies?

About Kidney Transplantation

*Even separating the process of kidney transplantation from the other two (heart and lung) as terminology is an important starting point for the recipients. Often, the term has changed from "deceased donor" to "living donor." As mentioned, the anxiety levels of the recipients here vary depending on the expression representing "the donor," such as living or deceased, intended or unintended donor.  *Do recipients receive support for ongoing psychological conditions? Is there a follow-up process in included references that draws your attention?

Author Response

Reviewer 1

I congratulate the authors on their topic. When we exclude the articles from 2021 in the included studies, we can see that these evaluations are made at intervals of more than ten years in the literature. Despite this time difference, their collection of such a critical subject shows the manuscript's importance. 

My minor comments:

*Inclusion and Exclusion criteria are well defined. 

*Limitations in literature and the lack of proper analysis are well compared.

Author’s response: We would like to thank you for these comments.

About Heart transplantation

*Developing new traits and resenting the loss of their own heart must have caused an emotional wave. However, according to the manuscript, most DDI observations start post-transplant. Did indignation about a possible loss of their own heart start pre-operatively? If possible, please indicate. 

Author’s response: We reviewed the publications in this regard once again. There are only two studies that report thoughts about losing one's heart, both only post-operatively.

We have added information about the post-operative observations in the respective studies., see p. 5: One patient in Stolf and Sadala's study [42] reported that he would always remember that he was not living with his own heart. Thoughts about losing one's heart are also hinted at in the study by Kuhn et al. [37]: A patient is quoted to have said that he/she would be leaving his/her heart in the hospital upon discharge. However, there is no further discussion of this or its impact.

Please also see "Time of occurrence" (p. 5): Thoughts about losing one's own heart were addressed by two studies [37,42]; in these, such thoughts occurred after transplantation. No other studies report such thoughts.

*All emotional processes related to the heart may be related to the patients' empathy feelings and the complex process they experience. In addition, although "cardiac memory" is a dilemma, it is still a current issue. Is there a justification for the acquired characteristic trait for cardiac memory? The manuscript mentioned the emotional adjustment process in the Kuhn 1988 study. However, brief information about cardiac memory can be added to the conclusion to increase the article's impact. (such as https://doi.org/10.1016/S0008-6363(98)00208-9).

Author’s response: Thank you very much for directing our attention to this aspect. We have included this aspect in our “discussion”, see p. 9: Physical memory processes, such as cardiac memory, might explain part of the modified self-perception in patients following heart transplantation. For instance, alterations in physiology such as regarding heart rhythm, stored in the donor heart (“remembered”), might bring about lingering changes in recipients´ physical perception of the organ [47], although this has to be considered highly speculative at this point. We have also included the suggested publication.

About Lung Transplantation

*The "a deceased donor" term influences many recipients' emotions about the donor. As reported in lung transplantation, regret reveals a complex focus of stress, such as refusal of possible emotional attachment to the donor (denial).  Is a standard mood assessment test preferred in the interview studies mentioned here? Was a standard method preferred for such comprehensive evaluations in those studies?

Author’s response: In the relevant study, semi-structured interviews were conducted. We have added this information accordingly (see p. 7: This information was not assessed by means of standardized survey instruments but emerged during semi-structured interviews).

We agree that the emotional experience of transplant recipients is complex. As we now describe in the discussion, “The type of questions used in the interview studies may have caused a bias (e.g., in terms of the emotional response of transplant recipients when referring to a “deceased” donor). This makes it all the more important to use standardized questionnaires” (see p. 9f.).

About Kidney Transplantation

*Even separating the process of kidney transplantation from the other two (heart and lung) as terminology is an important starting point for the recipients. Often, the term has changed from "deceased donor" to "living donor." As mentioned, the anxiety levels of the recipients here vary depending on the expression representing "the donor," such as living or deceased, intended or unintended donor.  *Do recipients receive support for ongoing psychological conditions? Is there a follow-up process in included references that draws your attention?

Author’s response: None of the papers report specific interventions related to DDI or differences in recipients´ follow-ups between living donations and donations from deceased individuals. However, this might be an exciting topic for future studies. We have included this aspect in our “discussion” (see p. 10: “Looking at differences in DDI and their psychological reflection between living donations and donations from deceased individuals represents an important topic for further studies”).

Reviewer 2 Report

This is a unique literature of DDI which is a less studied in transplantation. There are no prospective or large cohort study for the same. Hence this article is very relevant. 

Authors did an extensive literature search- including case reports and interview studies to look into DDI. They looked into various emotional changes which the recipients experience and how that impacts their post transplant quality of life. 

Our search revealed nine qualitative studies addressing the phenomenon of DDI in 123 heart transplantation; samples ranged from N = 14 to N = 47. Clarity is needed what N stands for? For instance number of subjects? If so  there is a case report from Kuhn where N = 65 (Table 1). 

Author Response

Reviewer 2

This is a unique literature of DDI which is a less studied in transplantation. There are no prospective or large cohort study for the same. Hence this article is very relevant. 

Authors did an extensive literature search- including case reports and interview studies to look into DDI. They looked into various emotional changes which the recipients experience and how that impacts their post transplant quality of life. 

Author’s response: We would like to thank you for these comments.

Our search revealed nine qualitative studies addressing the phenomenon of DDI in 123 heart transplantation; samples ranged from N = 14 to N = 47. Clarity is needed what N stands for? For instance number of subjects? If so there is a case report from Kuhn where N = 65 (Table 1). 

Author’s response: Thank you for this comment. We have changed the sentence accordingly: Our search revealed nine qualitative studies addressing the phenomenon of DDI in heart transplantation; samples ranged from N = 14 to N = 47 participants.

We agree that the indication of the people studied is somewhat confusing in Kuhn 1988. We have tried to clarify this in the table (see Table 1): “The authors mentioned having observed a total of 65 patients but describe six cases in their article.”
